

# Variations in length of stay among surviving preterm infants admitted to neonatal intensive care units in Shenzhen, China

Dandan Rao[1], Zhangbin Yu[2], Rongtian Liu[3], Rui Wang[1], Xin Guo[4], Huiying Tu[2], Ping Jiang[1], Jing Zhang[1], Jinxing Feng[5], Linying Yang[5], Yuqin Yan[2] and Jiebo Liu[1]

[1] Shenzhen People's Hospital (The Second Clinical Medical College, Jinan University; The First Affiliated Hospital, Southern University of Science and Technology), Shenzhen, Guangdong, China
[2] Department of Neonatology, Shenzhen People's Hospital, Shenzhen, Guangdong, China
[3] Department of Neonatology, Shenzhen Second People's Hospital, Shenzhen, Guangdong, China
[4] Department of Neonatology, Longgang District Maternity & Child Healthcare Hospital of Shenzhen City, Shenzhen, Guangdong, China
[5] Department of Neonatology, Shenzhen Children's Hospital, Shenzhen, Guangdong, China

Corresponding authors
Yuqin Yan, yanyuqing77@126.com
Jiebo Liu, 446107008@qq.com

## ABSTRACT

**Background**. The global preterm birth rate is currently around 10%, making it a major contributor to both neonatal mortality and long-term health complications. Length of stay (LOS) in hospital is a key metric when assessing the quality of medical care for preterm infants. Therefore, a comprehensive analysis of LOS in preterm infants, along with the identification of associated risk factors, is essential in order to improve clinical outcomes and optimize healthcare strategies.

**Objective**. This study examined the LOS and total hospitalization costs of preterm infants admitted to neonatal intensive care units (NICUs) in three hospitals in China . The study also investigated the factors affecting the LOS and total hospitalization costs of surviving preterm infants admitted to NICUs.

**Methods**. This is a retrospective analysis of preterm infants born between January 2020 and December 2023 who were admitted to one of three participating tertiary care centers within one week after birth. Baseline characteristics, LOS, and hospitalization costs were described and summarized. Generalized linear models were utilized to estimate adjusted associations between LOS and various factors.

**Results**. This study included 2,887 preterm infants. The median total LOS was 10 days (interquartile range (IQR): 7–20), and the median total hospitalization cost was RMB ¥16,287.3 (IQR: ¥10,541.9–32,342.3). Both LOS and hospitalization costs decreased significantly with increasing gestational age (GA) and birth weight (BW). Infants at 34 to 36+6weeks' gestation had a 68% shorter LOS compared to those at <28 weeks (relative risk (RR) 0.32, 95% confidence interval (CI) [0.27–0.38]). Similarly, infants with BW ≥ 2,500 g had a 54% shorter LOS than those <1,000 g (RR 0.46, 95% CI [0.39–0.54]).

**Conclusion**. Low GA, low BW, maternal hypertension, low Apgar score , small for GA (SGA), bronchopulmonary dysplasia (BPD), sepsis, and extrauterine growth retardation were associated with prolonged hospitalization.

## INTRODUCTION

There is growing global concern regarding preterm babies, as one in ten babies worldwide are born prematurely and the rate of preterm birth is on the rise (*Chawanpaiboon et al., 2019*). In China, over 1 million babies are born prematurely annually (*Zhang, Sun & Zhang, 2021*). Preterm birth is one of the leading causes of death in children under 5 years of age (*Liu et al., 2012*), and surviving preterm infants often face various complications such as neonatal bronchopulmonary dysplasia (BPD), necrotizing enterocolitis (NEC), and intraventricular hemorrhage (IVH), leading to extended hospital stays. The length of stay (LOS) is a key factor in determining costs and serves as an important measure for evaluating the quality of care.

From a financial standpoint, the hospitalization costs for preterm infants can be categorized into institutional costs (including hospital resource utilization and health insurance expenditures) and family-borne expenses (encompassing out-of-pocket payments and productivity losses due to caregiving) (*Russell et al., 2007*). Prolonged LOS not only decreases hospital bed availability and elevates the risk of healthcare-associated infections, but may also precipitate catastrophic health expenditures for households (*Xu et al., 2003*). Furthermore, extended parent-infant separation coupled with uncertainty regarding LOS duration significantly intensifies parental psychological distress (*Sandnes et al., 2024*). Consequently, precise prediction of both LOS and associated costs is crucial for enhancing resource allocation efficiency, facilitating better clinician-family communication, and enabling personalized discharge planning.

While there are established interventions for preterm infants, the lack of guidance on discharging these infants indicates a lack of consensus standards (*Maier et al., 2018*). Data from the iNeo network reveal a 3-week discrepancy in LOS for very preterm infants across different countries (*Seaton et al., 2021*), possibly due to differences in clinical care practices and healthcare systems. There have been few multicenter studies conducted on LOS in neonatal intensive care units in China (*Zhang et al., 2022*).

The LOS and cost of care for preterm infants can vary across regions due to disparities in medical, nursing, and economic resources. To offer valuable insights for healthcare professionals and parents, we conducted a study on the LOS and cost of preterm infants admitted to neonatal intensive care unit (NICU) in three hospitals in Shenzhen. Additionally, we investigated the factors influencing the LOS of preterm infants who survive their NICU stay.

## MATERIALS & METHODS

### Participating hospitals

This retrospective multicenter cohort study involved several hospitals, including Shenzhen People's Hospital and Shenzhen Luohu People's Hospital. All participating hospitals were

tertiary-level general hospitals with Class III NICUs, comprehensive medical resources, and advanced technical capabilities necessary to treat critically ill preterm infants.

## Study population

This study included preterm infants (gestational age (GA) less than 37 weeks) born between January 2020 and December 2023 who were admitted to one of three participating tertiary care centers within one week of birth. Exclusion criteria included significant congenital anomalies, discharge against medical advice, incomplete data, rehospitalization after initial discharge, transfer to a non-participating hospital, and infants who died prior to initial discharge. Data on infants transferred to participating hospitals were tracked until NICU discharge.

## Data collection

Detailed clinical data were extracted from the medical record system by trained data extractors. Data collected included maternal characteristics (maternal age, primiparity, maternal hypertension, maternal diabetes, Group B Streptococcus (GBS) infection, antenatal corticosteroids, cesarean section, and premature rupture of membranes (PROM) $\geq$ 18 h), infant characteristics (GA, birth weight (BW), male, small for GA (SGA), multiple birth, inborn infants, 1-minute Apgar scores $\leq$ 7, 5-minute Apgar scores $\leq$ 7, surfactant use, hospitalization costs and LOS), and major infant morbidities (IVH grade III and above or cystic periventricular leukomalacia (cPVL), NEC $\geq$ stage II, BPD, severe retinopathy of prematurity (ROP), sepsis, nosocomial infection, and extrauterine growth retardation).

## Definitions

GA was estimated and determined using prenatal ultrasound, menstrual history, obstetric examination, and Ballard score (*Ballard, Novak & Driver, 1979*). Extrauterine growth restriction (EUGR) was identified as a discharge weight falling below the 10th percentile for the same sex and GA based on the 2013 Fenton growth curve (*Fenton & Kim, 2013*). Small for GA (SGA) was defined as a BW below the 10th percentile for GA according to Chinese neonatal BW standards (*Zhu et al., 2015*). Antenatal corticosteroid use referred to mothers receiving at least one dose of dexamethasone or betamethasone before delivery. IVH was graded following the Papile criteria (*Papile et al., 1978*). cPVL was defined as the presence of periventricular cysts on cranial ultrasound or magnetic resonance imaging. Severe congenital malformations were those requiring immediate medical or surgical intervention. NEC of the small intestine was classified as stage II or higher per the Bell criteria (*Walsh & Kliegman, 1986*). Severe ROP was diagnosed based on the International Classification of Retinopathy of Prematurity (*Prematurity, 2005*). BPD was defined using the diagnostic criterion of requiring supplemental oxygen for $\geq$28 days after birth (*Jobe & Bancalari, 2001*). Sepsis encompassed clinical and definitive diagnosis. A clinical diagnosis required clinical abnormalities plus any one of the following: ① two or more positive non-specific blood tests, ② cerebrospinal fluid examination showing changes indicative of purulent meningitis, or ③ detection of pathogenic bacterial DNA in the blood. A definitive diagnosis required the presence of clinical manifestations and a positive blood

culture or cerebrospinal fluid (or other sterile luminal fluid) culture (*Subspecialty Group Of Neonatology & Professional Committee Of Infectious Diseases, 2019*).

## Statistical analysis

Categorical variables were compared using absolute numbers and percentages, as well as the $\chi 2$ test. Baseline characteristics, LOS, and hospital costs were summarized descriptively. Normally distributed quantitative baseline characteristics were presented as means with standard deviations, while highly skewed variables were expressed as medians with interquartile ranges. Multiple group comparisons of the baseline characteristics table were performed using the Kruskal–Wallis test and the chi-squared test.

Box plots were employed to visualize the distributional relationships of GA and BW with LOS and hospitalization costs. We employed multiple imputation to handle missing data. Both LOS and hospitalization costs showed right-skewed distributions that remained non-normal after log-transformation. Consequently, generalized linear models (GLMs) with a gamma distribution were selected to determine the associations of perinatal factors and neonatal complications with LOS and hospitalization costs. Variables showing a significant association with LOS (or hospitalization costs) ($P < 0.05$) in the univariate analysis (Model 1) were included in subsequent multivariate regression analyses. Model 2 adjusted for significant maternal and neonatal characteristics identified in Model 1. Model 3 adjusted for significant major infant morbidities identified in Model 1. Model 4 adjusted for all significant factors identified in Model 1. In the multivariate analyses, results with a $P$-value $< 0.05$ were considered statistically significant.

Statistical analyses were performed using IBM SPSS statistical software (version 27.0), while graphical visualization analysis was performed using R 4.2.1 software. The results of the study were expressed as significant at $P < 0.05$.

## Ethical approval

This study was conducted in compliance with the Declaration of Helsinki and received ethical approval from the Shenzhen People's Hospital (approval No. LL-KY-2024116-01), which was also acknowledged by all participating hospitals. As a retrospective study utilizing deidentified data, the requirement for informed consent was waived by the ethics committee.

## RESULTS

During the study period, a total of 3,329 preterm infants were admitted to the NICU within 7 days of birth at the three participating hospitals. Of these, 38 had major congenital anomalies, 295 were discharged against medical advice, four had incomplete data, 25 were readmitted after their first discharge, 64 were transferred to other non-participating hospitals for any reason, and 16 died prior to their first discharge. All these cases were excluded and the final number of preterm infants included in the study was 2,887.

Table 1 provides detailed baseline characteristics of the study population. The enrolled preterm infants had a median GA of 35.3 (33.7–36.1) weeks, a median BW of 2,300 (1,900–2,630) g, a median LOS of 10 (7–20) days, and a median hospitalization cost of

**Table 1  Baseline characteristics of preterm infants surviving to discharge from the neonatal intensive care unit in Shenzhen, China.** Categorical variables were compared using absolute numbers and percentages, as well as the $\chi^2$ test. Baseline characteristics, LOS, and hospital costs were summarized descriptively. Normally distributed quantitative baseline characteristics were presented as means with standard deviations, while highly skewed variables were expressed as medians with interquartile ranges. Multiple group comparisons of the baseline characteristics table were performed using the Kruskal–Wallis test and the chi-squared test.

| Characteristics | Total ($n = 2,887$) | <28wk ($n = 82$) | 28–31+6wk ($n = 298$) | 32–33+6wk ($n = 413$) | 34–36+6wk ($n = 2094$) | P |
|---|---|---|---|---|---|---|
| **Maternal characteristics** | | | | | | |
| Maternal age(y), median(IQR) | 31 (29–35) | 33 (29–36) | 32 (28–35) | 31 (28–35) | 32 (29–35) | 0.186 |
| Primigravida, n/N(%) | 1,546/2,887 (53.6) | 45/82 (54.9) | 155/298 (52.0) | 242/413 (58.5) | 1,104/2,094 (52.7) | <0.001 |
| Maternal hypertension, n/N(%) | 495/2,886 (17.2) | 10/82 (12.2) | 86/298 (28.8) | 96/413 (23.2) | 303/2,093 (14.5) | <0.001 |
| Maternal diabetes, n/N(%) | 735/2,886 (25.5) | 23/82 (28.0) | 80/298 (26.8) | 102/413 (24.7) | 530/2,093 (25.3) | 0.865 |
| GBS infection, n/N(%) | 141/2,771 (5.1) | 5/75 (6.7) | 9/271 (3.3) | 22/384 (5.7) | 105/2,041 (5.1) | 0.477 |
| Antenatal corticosteroids, n/N(%) | 1,056/2,887 (36.6) | 58/82 (70.7) | 203/298 (68.1) | 252/413 (61.0) | 543/2,094 (25.9) | <0.001 |
| cesarean section, n/N(%) | 2,148/2,887 (74.4) | 40/82 (48.8) | 243/298 (81.5) | 334/413 (80.8) | 1,531/2,094 (73.1) | <0.001 |
| PROM $geq$ 18 hours, n/N(%) | 420/2,879 (14.6) | 19/79 (24.1) | 77/297 (25.9) | 81/413 (19.6) | 243/2,090 (11.6) | <0.001 |
| **Infant characteristics** | | | | | | |
| GA (wk), median (IQR) | 35.3 (33.7–36.1) | 27.1 (26.0–27.4) | 30.2 (29.3–31.1) | 33.1 (32.6–33.6) | 35.9 (35.0–36.3) | |
| Birth weight (kg), median (IQR) | 2,300 (1,900–2,630) | 935 (770–1,072) | 1,330 (1,150–1,520) | 1,870 (1,660–2,090) | 2,490 (2,210–2,720) | <0.001 |
| SGA, n/N (%) | 299/2,887 (10.4) | 6/82 (7.3) | 31/298 (10.4) | 42/413 (10.1) | 220/2,094 (10.5) | 0.834 |
| Male, n/N (%) | 1,608/2,887 (55.7) | 46/82 (56.1) | 170/298 (57.0) | 198/413 (47.9) | 1,194/2,094 (57.0) | 0.008 |
| Multiple birth, n/N (%) | 811/2,887 (28.1) | 31/82 (37.8) | 81/298 (27.1) | 135/413 (32.6) | 564/2,094 (26.9) | 0.021 |
| Inborn, n/N (%) | 2,865/2,887 (99.2) | 79/82 (96.3) | 293/298 (98.3) | 411/413 (99.5) | 2,082/2,094 (99.4) | |
| Apgar score $\leq$ 7 at 1 min, n/N (%) | 161/2,883 (5.6) | 37/80 (46.3) | 56/297 (18.9) | 27/413 (6.5) | 41/2,093 (2.0) | <0.001 |
| Apgar score $\leq$ 7 at 5 min, n/N (%) | 33/2,883 (1.1) | 12/80 (15.0) | 11/297 (3.7) | 3/413 (0.7) | 7/2,093 (0.3) | <0.001 |
| Surfactant use, n/N (%) | 399/2,887 (13.8) | 73/82 (89.0) | 181/298 (60.7) | 79/413 (19.1) | 66/2,094 (3.1) | <0.001 |
| Hospitalization costs (RMB), median (IQR) | 16,287.3 (10,541.9–32,342.3) | 155,269.0 (126,984.8–195,004.9) | 87,685.8 (65,954.0–108,497.3) | 35,005.9 (23,660.7–49,433.3) | 13,248.0 (8503.1–1,829.0) | <0.001 |
| LOS (d), Median (IQR) | 10.0 (7.0–20.0) | 78.0 (68.0–87.3) | 47.5 (36.0–57.3) | 23.0 (16.0–29.0) | 8.0 (6.0–11.0) | <0.001 |
| **Major infant morbidities** | | | | | | |
| IVH grade III and above or cPVL, n/N (%)[a] | 29/2,850 (1.0) | 8/82 (9.8) | 9/298 (3.0) | 4/413 (0.9) | 8/2,057 (0.4) | <0.001 |
| NEC $\geq$ stage II, n/N(%) | 19/2,887(0.6) | 1/82 (1.2) | 12/298 (4.0) | 2/413 (0.4) | 4/2094 (0.1) | <0.001 |
| BPD, n/N(%) | 126/2,887 (4.4) | 66/82 (80.5) | 55/298 (18.4) | 5/413 (1.2) | 0/2,094 (0) | <0.001 |
| Severe ROP, n/N(%)[b] | 16/2,065 (0.8) | 10/82 (12.2) | 3/291 (1.0) | 1/318 (0.3) | 2/1,374 (0.1) | <0.001 |
| Sepsis, n/N(%) | 74/2,887 (2.6) | 17/82 (20.7) | 34/298 (11.4) | 8/413 (1.9) | 15/2,094 (0.7) | <0.001 |
| Nosocomial infection, n/N(%) | 263/2,887 (9.1) | 41/82 (50.0) | 69/298 (23.1) | 34/413 (8.2) | 119/2,094 (5.6) | <0.001 |
| EUGR, n/N(%) | 752/2,805 (26.8) | 23/81 (28.4) | 99/290 (34.1) | 109/398 (27.4) | 521/2,036 (25.6) | 0.022 |

**Notes.**

GBS, Group B Streptococcus; PROM, premature rupture of membranes; LOS, length of stay; GA, gestational age; IQR, interquartile range; SGA, small for gestational age; IVH, intraventricular hemorrhage; cPVL, cystic periventricular leukomalacia; NEC, necrotizing enterocolitis; BPD, bronchopulmonary dysplasia; ROP, retinopathy of prematurity; EUGR, extrauterine growth restriction

[a]Incidence of IVH grade III and above or cPVL was calculated within infants who had neuroimaging results.

[b]Incidence of ROP was calculated within infants who finished the ROP screening.

¥16,287.3 (¥10,541.9–32,342.3) RMB. A higher proportion of primiparous (53.6%), cesarean (74.4%), and male (55.7%) births were observed among these preterm infants. As GA decreased, there was a progressive increase in the incidence of IVH grade III and above or cPVL, NEC $\geq$ stage II, BPD, severe ROP, sepsis in surviving preterm infants, as

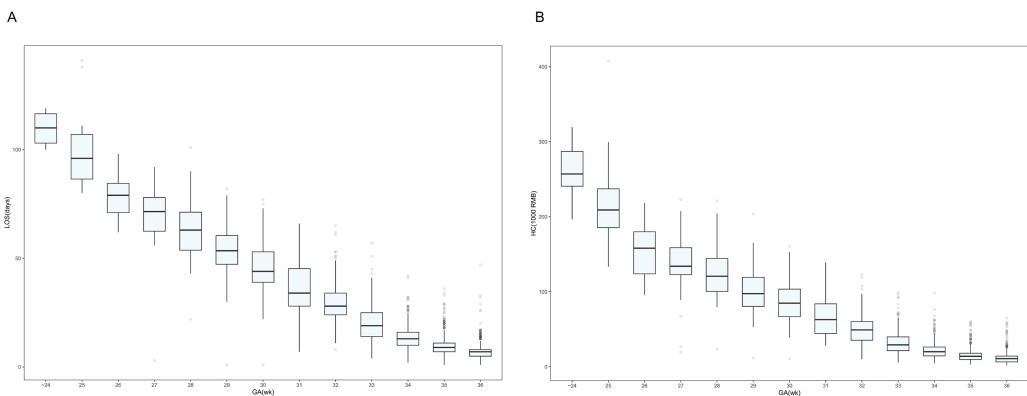

**Figure 1   Box plots showing the correlation between gestational age and length of stay and hospitalization costs.** (A) Distribution of hospitalization length by gestational age of surviving preterm infants in Shenzhen NICUs. (B) Distribution of hospitalization costs by gestational age of surviving preterm infants in Shenzhen NICUs. Note: lOS, length of stay; GA, gestational age; HC, hospitalization costs.

well as nosocomial infection. In multiple group comparisons of baseline characteristics, significant differences ($p < 0.05$) were observed in maternal factors (primigravida, maternal hypertension, multiple birth), obstetric factors (antenatal corticosteroids, cesarean section, PROM ≥18 h), neonatal characteristics (BW, male, Apgar scores ≤7 at 1 and 5 min), interventions (surfactant use), outcomes (IVH grade III and above or cPVL, NEC ≥ stage II, BPD, severe ROP, sepsis, nosocomial infection, EUGR), and healthcare utilization (hospitalization costs, LOS).

Figures 1A and 1B show, by GA, the distribution of LOS and hospitalization costs, respectively. Visual analysis of the box plots indicates that the median LOS and hospitalization costs generally increase with decreasing GA. Infants born at a GA of 24 weeks or earlier had the longest median LOS (110 days) and the highest median hospitalization cost (RMB ¥257,051.6), while those born at 36 weeks had the shortest median LOS (7 days) and the lowest median hospitalization cost (RMB ¥10,806.3). Notably, the mean GA at discharge decreased from 40.3 weeks in the ≤24 weeks group to approximately 36 weeks in the 33-36 weeks group. Complete data for each GA subgroup are shown in Table S1.

The lower the BW, the higher the LOS and hospitalization costs (Figs. 2A and 2B). Depending on the infant's BW, the median LOS was 77 days for BW <1,000 g, 57 days for BW 1,000–1,249 g, 43 days for BW 1,250–1,499 g, 12 days for BW 1,500–2,499 g, and 7 days for BW ≥ 2,500 g (Table S1).

Table 2 and Table S2 present factors associated with LOS and hospitalization costs. Univariate analysis identified several factors linked to increased LOS and costs: maternal hypertension, antenatal corticosteroid use, PROM ≥ 18 h, lower GA, lower BW, SGA, multiple birth, Apgar score ≤7 at 1 and 5 min, surfactant use, IVH grade III and above or cPVL, NEC ≥ stage II, BPD, severe ROP, nosocomial infection, sepsis, and EUGR. Primiparity was only associated with prolonged LOS. Model 2, adjusted for maternal and neonatal characteristics, showed that maternal hypertension, antenatal corticosteroid use, lower GA, lower BW, SGA, Apgar score ≤7 at 1 min, and surfactant use remained

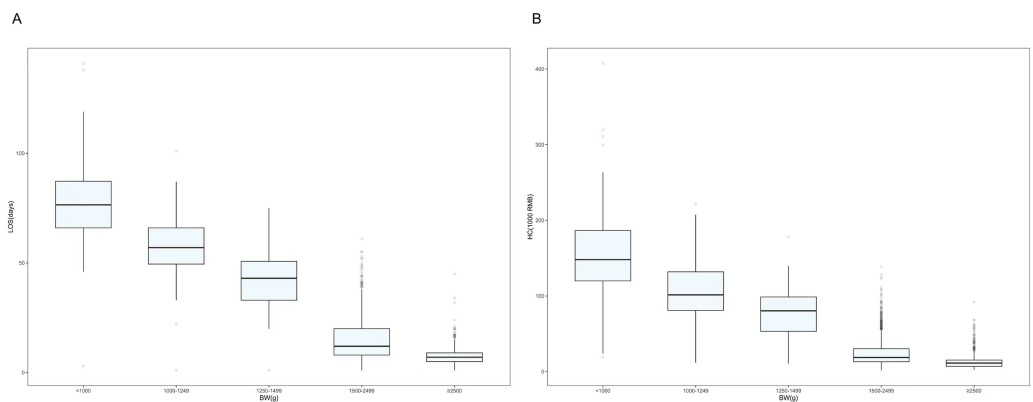

**Figure 2  Box plots showing weight in relation to length of stay and hospitalization costs.** (A) Distribution of hospital length of stay by weight of surviving preterm infants in Shenzhen NICUs. (B) Distribution of hospitalization costs by weight of surviving preterm infants in Shenzhen NICUs. Note: LOS, length of stay; BW, Birth weight; HC, hospitalization costs.

significantly associated with both prolonged LOS and increased costs. Multiple births were only associated with higher costs. Model 3, adjusted for major neonatal morbidities, indicated that each major morbidity type was positively associated with increased LOS and costs. Model 4, which incorporated all significant factors from Model 1, yielded results for maternal and neonatal factors consistent with Model 2. Higher GA and BW were significantly associated with reduced LOS and costs. Compared to infants with GA <28 weeks, those at 34–36$^{+}$6 weeks had a 68% shorter LOS (relative risk (RR) 0.32, 95% confidence interval (CI) [0.27–0.38]). Similarly, infants with BW ≥2,500 g had a 54% shorter LOS compared to those <1,000 g (RR 0.46, 95% CI [0.39–0.54]). After adjustment, BPD (RR 1.13, 95% CI [1.02–1.26]), sepsis (RR 1.13, 95% CI [1.02–1.26]), and EUGR (RR 1.09, 95% CI [1.04–1.14]) remained significantly associated with prolonged LOS.

## DISCUSSION

This retrospective multicenter cohort study aimed to analyze the LOS and total hospitalization cost of preterm infants discharged from the NICU in three hospitals in Shenzhen, China. The study also sought to identify factors influencing the LOS of surviving preterm infants. The findings indicated that LOS and hospitalization costs decreased as GA and BW increased.

In our study, the median LOS for preterm infants was 10 days. Our findings regarding LOS for very preterm infants align with a multicenter study conducted in Chinese NICUs (*Zhang et al., 2022*). Seaton et al. found that LOS for infants with a GA of 29 to 31 weeks was similar to our results, while infants born at ≤ 28 weeks had a hospitalization length approximately 10 days longer than our findings (*Seaton et al., 2019*). Another study from the iNeo network reported that the median LOS for surviving 24- to 28-week-old infants varied from 75 days in Finland to 107 days in Japan, with an overall median LOS of 87 days (*Seaton et al., 2021*). In our study, the median LOS for infants <28 weeks was 78 days, which is 9 days less compared to developed countries. This difference may be attributed

**Table 2  Factors affecting the length of stay of surviving preterm infants in neonatal intensive care units in Shenzhen, China.** Using generalized linear models (GLMs) with gamma distribution to determine the association between perinatal factors and neonatal complications with length of stay (LOS).

| Factors | Model 1 | Model 2 | Model 3 | Model 4 |
|---|---|---|---|---|
| | RR (95% CI) | RR (95% CI) | RR (95% CI) | RR (95% CI) |
| **Maternal characteristics** | | | | |
| Maternal age | 1.00 (0.99, 1.01) | | | |
| Primigravida | **1.07 (1.01, 1.14)** | 1.01 (0.98, 1.04) | | 1.00 (0.97, 1.04) |
| Maternal hypertension | **1.47 (1.35, 1.59)** | **1.12 (1.07, 1.17)** | | **1.12 (1.07, 1.17)** |
| Maternal diabetes | 1.01 (0.94, 1.08) | | | |
| GBS infection | 0.94 (0.82, 1.09) | | | |
| Antenatal corticosteroids | **1.91 (1.79, 2.02)** | **1.12 (1.09, 1.16)** | | **1.13 (1.09, 1.17)** |
| cesarean section | 1.07 (0.99, 1.14) | | | |
| PROM ≥ 18 h | **1.25 (1.14, 1.36)** | 1.03 (0.99, 1.08) | | 1.03 (0,98, 1.08) |
| **Infant characteristics** | | | | |
| GA(wk) | | | | |
| <28 | Reference | Reference | | Reference |
| 28–31$^{+6}$ | **0.59 (0.53, 0.67)** | **0.79 (0.70, 0.90)** | | **0.83 (0.72, 0.95)** |
| 32–33$^{+6}$ | **0.30 (0.26, 0.33)** | **0.59 (0.51, 0.69)** | | **0.62 (0.53, 0.74)** |
| 34–36$^{+6}$ | **0.11 (0.10, 0.13)** | **0.31 (0.26, 0.36)** | | **0.32 (0.27, 0.38)** |
| Birth weight (g) | | | | |
| <1,000 | Reference | Reference | | Reference |
| 1,000–1,249 | **0.73 (0.63, 0.84)** | 0.93 (0.81,1.05) | | 1.02 (0.89, 1.17) |
| 1,250–1,499 | **0.54 (0.47, 0.62)** | **0.80 (0.70, 0.92)** | | **0.91 (0.79, 1.05)** |
| 1,500–2,499 | **0.19 (0.17, 0.21)** | **0.56 (0.49, 0.65)** | | **0.65 (0.56, 0.76)** |
| ≥ 2,500 | **0.09 (0.08, 0.10)** | **0.38 (0.33, 0.45)** | | **0.46 (0.39, 0.54)** |
| SGA | **1.41 (1.28, 1.56)** | **1.19 (1.12, 1.26)** | | **1.13 (1.07, 1.20)** |
| Male | **0.92 (0.86, 0.97)** | 1.00 (0.97, 1.03) | | 0.99 (0.97, 1.03) |
| Multiple birth | **1.12 (1.05, 1.2)** | 0.99 (0.96, 1.03) | | 0.99 (0.96, 1.03) |
| Apgar score ≤ 7 at 1 min | **2.90 (2.56, 3.28)** | **1.1 (1.02, 1.19)** | | **1.08 (1.005, 1.17)** |
| Apgar score ≤ 7 at 5 min | **2.99 (2.26, 3.97)** | 1.04 (0.89, 1.21) | | 1.08 (0.92, 1.28) |
| surfactant use | **3.52 (3.27, 3.79)** | **1.23 (1.17, 1.30)** | | **1.22 (1.15, 1.29)** |
| **Major infant morbidities** | | | | |
| Nosocomial infection | **2.32 (2.10, 2.57)** | | **1.36 (1.23, 1.49)** | 0.99 (0.94, 1.06) |
| IVH grade III and above or cPVL | **2.64 (1.95, 3.58)** | | **1.56 (1.19, 2.02)** | 1.07 (0.91, 1.25) |
| NEC ≥ stage II | **2.72 (1.87, 3.95)** | | **1.94 (1.41, 2.67)** | 1.2 (0.99, 1.46) |
| BPD | **4.90 (4.29, 5.59)** | | **3.78 (3.31, 4.32)** | **1.13 (1.02, 1.26)** |
| Severe ROP | **4.82 (3.22, 7.22)** | | **1.47 (1.01, 2.12)** | 1.26(1.00, 1.57) |
| Sepsis | **3.23 (2.68, 3.90)** | | **1.88 (1.59, 2.23)** | **1.28 (1.16, 1.42)** |
| EUGR | **1.47 (1.37, 1.57)** | | **1.38 (1.30, 1.47)** | **1.09 (1.04, 1.14)** |

**Notes.**

GBS, Group B Streptococcus; PROM, premature rupture of membranes; LOS, length of stay; GA gestational age; SGA, small for gestational age; IVH intraventricular hemorrhage, cPVL, cystic periventricular leukomalacia; NEC, necrotizing enterocolitis; BPD, bronchopulmonary dysplasia, ROP, retinopathy of prematurity; EUGR, extrauterine growth restriction; RR, relative risk. Model 1: Crude. Model 2: Adjust: primigravida, maternal hypertension, antenatal corticosteroids, gestational age, birth weight, small for gestational age, infant sex, multiple birth, surfactant use, Apgar score ≤ 7 at 1 min, Apgar score ≤ 7 at 5 min, and preterm rupture of membranes ≥ 18 h. Model 3: Adjust: nosocomial infections, IVH grade III and above or cPVL, NEC ≥ stage II, BPD, severe ROP, sepsis, and EUGR. Model 4: Adjust Model 2 plus Model 3. Significant results are in bold.

to higher survival rates in developed countries, especially in the youngest GA infants who are at higher risk of complications and longer hospital stays. A multicenter study in India showed that LOS for infants born between 25–33 weeks GA, as well as median LOS and postmenstrual age (PMA) at discharge, were lower compared to our study. Infants in India were discharged earlier in all BW categories, which may be due to variations in discharge criteria in different countries and regions. Additionally, infants in India were transferred to Kangaroo Mother Care (KMC) wards as soon as they were stable (*Murki et al., 2020*), which has been shown to not only reduce LOS but also alleviate financial burdens on parents (*Sharma, Murki & Oleti, 2018*). Although KMC was not directly addressed in this study, its role in reducing LOS and costs has been widely recognized. Future research could explore the interaction between KMC and the correlates found in this study (*e.g.*, low BW, BPD).

Previous studies have found that the cost of neonatal intensive care varies widely across health systems. Rolnitsky et al. reported median hospitalization costs for extremely preterm infants (less than 28 weeks) in Canada to be $66,669 (2011–2015) and $77,132 (2010–2017), respectively (*Rolnitsky et al., 2023*; *Rolnitsky et al., 2021*). In contrast, costs were much higher in the U.S., where Beam et al. reported average costs to be $291,029 for 27- to 28-week-old infants, $418,191 for 25- to 26-week-old infants, and a sharp decline to $9,864 for 35- to 36-week-old infants (2008–2016) (*Beam et al., 2020*). These costs have risen in recent years. In Asia, the cost of hospitalization for extremely preterm infants in South Korea is approximately $29,101 (*Lee et al., 2025*). Meanwhile, the median hospitalization cost in this study (RMB ¥155,269) was significantly lower than that in North America and South Korea. This is consistent with the Fang et al. study that reported a cost of ¥152,979 RMB in 2019 (*Fang, Xu & Dai, 2021*). The intensive care costs for extremely preterm infants were six times higher than those for late preterm infants and 25 times higher than the minimum level of care. These findings are consistent with those of *Zainal et al. (2019)*. The high medical costs associated with extremely preterm infants can put significant financial strain on families, particularly when regional differences in health insurance reimbursement rates exist. This burden is further compounded by non-medical and indirect costs, as evidenced in a European study where families reported substantial expenses related to NICU hospitalization and post-discharge care (*Lambiase et al., 2024*). Some families are forced to discharge their babies early due to financial difficulties. This increases the risk of readmission and affects the babies' long-term prognosis. Given the growing burden of caring for premature babies, there is an urgent need for differentiated health policies to address this social health issue.

Research indicates that lower GA and BW are associated with prolonged hospitalization and increased costs, with infants born at <24 weeks gestation experiencing the longest mean hospital stay (110 days) and highest mean cost (¥257,051.6 RMB), primarily due to underdeveloped organ systems leading to heightened complication risks and aggravated conditions (*Russell et al., 2007*; *Phibbs et al., 2019*). Consequently, reducing preterm birth rates and associated complications, as well as shortening hospital stays, could significantly decrease medical expenditures and alleviate family financial burdens. Key strategies used to achieve this include enhancing healthcare provider training to improve antenatal care

quality and awareness for timely management of pregnancy risks, optimizing delivery methods and neonatal resuscitation skills, implementing specialized nursing techniques and individualized in-hospital care to reduce complications and mortality (*Mahwasane et al., 2020*), and actively promoting breastfeeding within the NICU, particularly for extremely preterm infants, to mitigate disease severity and costs (*Johnson et al., 2014*). Furthermore, effective cost management strategies like clinical pathways (*Askari, Tam & Klundert, 2021*), diagnosis-related groups (DRGs), or direct insurer payment (DIP) should be employed. Careful discharge planning, including KMC to strengthen bonding and reduce anxiety, along with structured follow-up, is essential to prevent premature discharge and readmissions.

Baseline characterization of preterm infants showed that the prevalence of severe complications in preterm infants increased with decreasing GA, which is consistent with previous studies (*Zhang et al., 2022*; *Thébaud et al., 2019*; *Bajwa et al., 2011*; *Dammann, Hartnett & Stahl, 2023*). However, the present study found that GA <28 weeks (1.2%) had a lower incidence of NEC $\geq$ II compared with 28–31$^{+6}$ weeks (4.0%). This was because none of the three hospitals were equipped for surgical procedures. Extremely preterm infants <28 weeks' gestation who developed NEC $\geq$ II were more severely ill, with a higher risk of perforation, and often requiring surgical intervention. Consequently, many of these patients were transferred to higher level hospitals, leading to a lower incidence of NEC $\geq$ II in those retained in the hospital. Additionally, previous studies have shown that children who survive are at a higher risk for cerebral palsy, visual and hearing impairments, respiratory complications, motor impairments, cognitive impairments, and mental health problems than children who are delivered at term (*Petrou, Yiu & Kwon, 2019*; *Collaborators, 2024*).

Our study confirmed that GA and BW were the strongest determinants of both LOS and hospitalization costs. Infants born at <28 weeks GA incurred nearly triple the LOS and costs compared to those born at 34-36 weeks, consistent with prior reports (*Murki et al., 2020*; *Xie et al., 2022*). Antenatal and perinatal factors, including maternal hypertension, 1-minute Apgar score $\leq$7, and SGA, were also associated with prolonged LOS, aligning with known literature (*Zhang et al., 2022*; *Pepler, Uys & Nel, 2012*). Neonatal complications such as BPD, sepsis, and EUGR further increased LOS and costs, which was as expected and has been widely reported (*Zhang et al., 2022*; *Murki et al., 2020*; *Xie et al., 2022*; *Kurihara, Zhang & Mikhael, 2021*). Therefore, as survival rates for very preterm infants improve, the clinical focus should shift from survival alone to achieving complication-free survival. This shift can significantly reduce NICU LOS, lower healthcare costs, and improve long-term outcomes. Notably, IVH grade III and above or cPVL showed no significant association with LOS or costs in our cohort. This lack of association likely stems from the absence of cases requiring surgical intervention or complications like hydrocephalus necessitating prolonged hospitalization before discharge. Conversely, both antenatal corticosteroid and postnatal surfactant administration were associated with increased LOS and costs in adjusted and unadjusted analyses. We propose this association is confounded by lower GA typical of infants receiving these interventions, rather than indicating these treatments themselves as independent risk factors for prolonged LOS or higher costs.

This study presents the first multicenter survey on LOS and hospitalization costs of preterm infants in Shenzhen, providing valuable insights for medical professionals and parents. However, there are several limitations to consider. First, this study did not treat preterm labor as a heterogeneous syndrome with different etiologies. In other words, it did not consider phenotypic differences in preterm births but rather only categorized them by gestational week. However, these phenotypic differences have been shown to significantly affect neonatal prognosis. Therefore, future studies should consider these factors to provide a more precise analysis (*Villar et al., 2024*). Second, the survey was limited to three tertiary hospitals in Shenzhen. Due to the limited sample size, future studies should include a larger number of hospitals to ensure better representation of the entire urban area. Third, discharge criteria varies from hospital to hospital and may affect the outcome of LOS. Finally, due to the limitations of retrospective studies, causal inferences were restricted, even though the findings suggested strong associations between these factors and increased length of hospitalization and costs. These factors may be markers or intermediate variables of complex pathological processes rather than direct causes. Unmeasured confounders, such as healthcare resource availability and quality (*Qattea et al., 2024*), socioeconomic status, and hospital management strategies, may influence both complications and LOS, further complicating causal interpretations. Future studies should use more rigorous designs to explore the mechanisms and true impact of these factors on healthcare resources.

## CONCLUSION

The results showed that low GA, low BW, maternal hypertension, SGA, BPD, sepsis, and EUGR were associated with prolonged hospitalization and increased costs. Future research should focus on comprehensive prospective studies to explore risk factors influencing LOS and hospitalization costs in preterm infants. Additionally, machine learning techniques could be utilized to develop predictive models, improving the accuracy of outcome predictions across various disease states.

### Funding
The authors received no funding for this work.

### Competing Interests
The authors declare there are no competing interests.

### Author Contributions
- Dandan Rao conceived and designed the experiments, performed the experiments, analyzed the data, prepared figures and/or tables, authored or reviewed drafts of the article, and approved the final draft.
- Zhangbin Yu conceived and designed the experiments, analyzed the data, authored or reviewed drafts of the article, and approved the final draft.

- Rongtian Liu performed the experiments, analyzed the data, prepared figures and/or tables, and approved the final draft.
- Rui Wang analyzed the data, prepared figures and/or tables, and approved the final draft.
- Xin Guo performed the experiments, analyzed the data, prepared figures and/or tables, and approved the final draft.
- Huiying Tu performed the experiments, prepared figures and/or tables, and approved the final draft.
- Ping Jiang performed the experiments, analyzed the data, prepared figures and/or tables, and approved the final draft.
- Jing Zhang performed the experiments, prepared figures and/or tables, and approved the final draft.
- Jinxing Feng conceived and designed the experiments, analyzed the data, prepared figures and/or tables, and approved the final draft.
- Linying Yang analyzed the data, prepared figures and/or tables, and approved the final draft.
- Yuqin Yan conceived and designed the experiments, analyzed the data, authored or reviewed drafts of the article, and approved the final draft.
- Jiebo Liu conceived and designed the experiments, analyzed the data, authored or reviewed drafts of the article, and approved the final draft.

## Human Ethics

The following information was supplied relating to ethical approvals (*i.e.*, approving body and any reference numbers):

Shenzhen People's Hospital approved the study (Approval No. LL-KY-2024116-01)

## Ethics

The following information was supplied relating to ethical approvals (i.e., approving body and any reference numbers):

This study was conducted in compliance with the Declaration of Helsinki and received ethical approval from the Shenzhen People's Hospital (Approval No. LL-KY-2024116-01), which was also acknowledged by all participating hospitals.

## Data Availability

The raw data are available in the Supplemental Files.

## Supplemental Information

Supplemental information for this article can be found online at http://dx.doi.org/10.7717/peerj.20344#supplemental-information.

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
