# Peer review of "Variations in length of stay among surviving preterm infants admitted to neonatal intensive care units in Shenzhen, China"

_PeerJ, doi:10.7717/peerj.20344_

## Round 0.1 · original submission · Major Revisions

· Academic Editor

Major Revisions

**Language Note:** The review process has identified that the English language must be improved. PeerJ can provide language editing services - please contact us at [email protected] for pricing (be sure to provide your manuscript number and title). Alternatively, you should make your own arrangements to improve the language quality and provide details in your response letter. – PeerJ Staff

Reviewer 1 ·

Basic reporting

-

Experimental design

Consideration of PTB Phenotypes
The manuscript currently treats PTB as a relatively homogenous entity categorized primarily by gestational age brackets. However, it is now well-established that PTB is a heterogeneous syndrome with varying underlying etiologies, such as infection/inflammation, maternal vascular disease, placental dysfunction, or fetal distress. These phenotypic differences can significantly influence neonatal outcomes. The recent work by Villar et al. (Villar J, et al. Etiologically Based Functional Taxonomy of the Preterm Birth Syndrome. Clin Perinatol. 2024 Jun;51(2):475-495. doi: 10.1016/j.clp.2024.02.014) proposes a clinically meaningful taxonomy of PTB phenotypes that could be integrated into outcome analyses. Including such stratification could uncover differential risks and better inform clinical decision-making. At minimum, the limitations section should acknowledge the absence of PTB phenotype data and suggest it as a key consideration for future research.

Validity of the findings

This is an interesting manuscript with important clinical relevance. The authors have handled a complex topic with clarity and have successfully synthesized a large dataset to generate meaningful insights. The inclusion of granular gestational age categories (e.g., extremely, very, moderate, and late preterm) enhances the practical utility of the findings. The paper is well-written, and the figures and tables are informative and appropriately placed.

They should do an effect going beyond the definition of PTB based on the GA categories. Please look at the PTB phenotypes.

Additional comments

Confounding Variables and Residual Confounding
Although the authors adjust for a range of covariates, it remains unclear whether all relevant maternal and perinatal factors (e.g., intrauterine growth restriction, hypertensive disorders) were included. Some of these are not only confounders but also potential mediators. Clarifying the rationale for variable selection would improve the transparency of the analysis.

Longitudinal Outcomes
While the study focuses on neonatal outcomes, the implications of early life complications often extend into childhood and adulthood. The authors might briefly discuss the trajectory of these morbidities beyond the neonatal period, even if outside the scope of the current dataset.

Presentation of Risk Estimates
Consider presenting both absolute risks and relative risks in the main text or supplementary material. This would help clinicians better contextualize the findings.

·

Basic reporting

Thank you for the opportunity to review this interesting manuscript.

- I suggest to review the english written form of the paper throughout the all text as there are several sentences that are not clear and ther are some typos as well. Some of them are the following one: line 48-preterm infants born in January 2020 to discharged in December 2023-. It is not clear if you included or not infant who were born from January 2020 to December 2023 . Another example could be lines 127-130 where is not clear if the presence or the absence of the three criteria lead to the diagnosis of sepsis.

-I recommend to expand the number of literature references used. In particular, within the introduction section you need to briefly report what is financial burden, from which perspective you are going to measure it (e.g. Instituion or Patient perspective) how and which aspect they will measure (i.e. subjective vs. objective measure). Then I would suggest to focus on studies out of China that examined the financial burden experienced within the NICU context. Finally, I would focus on the financial burden within the NICU context in China.
As regard to the discussion section, you also need to increase the number of references used to compare your findings both within Country and between Countries as costs could widely depend on the healthcare context of each region and of each Country as well.

- You can talk about the related parental worry and anxiety related to hospitalization and financial issues but you need to provide valid references.

Experimental design

I would suggest to improve method description.
In particular, I suggest to always state the type of statistical test and the aim that you have using that type of test. You need to make clear the research questions and the way you want to answer to each question specifying the test you want to use. For instance, the way and the order you insert variables in the regression model should be related to answer your research question.

Validity of the findings

I recommend to carefully check for the text as you used the term "correlations" to refer to regression models.
I suggest to revise figures as the box plots are not the perfect way to visualize correlations. Morevover regression table is hard to read. I would suggest to split it up according to the outcome (LOS or costs) and making statistically significant results more evident (e.g., underying them in bold). Moreover, when you talk about statistical results in the text is better to indicate the values of the statistical test that you used.
Line 180-186: these lines are very hard to read as you put all the gestational ages and costs. if you cannot use the table of the supplementary material I would find a better way to sinthesize the main findings of that table in the text.

Additional comments

In the method section, you should provide the level of the Neonatal Intensive Care Units involved in the study.

---

## Round 0.2 · Minor Revisions

· Academic Editor

Minor Revisions

·

Basic reporting

The manuscript is improved. The authors followed all the suggestions provided carefully.

Experimental design

Research questions have been stated clearly, and the methods used to answer are well explained.

Validity of the findings

I got the point to show the distribution using a box plot. However, when authors state:

"(lines 173): "A lower GA at birth is associated with a longer LOS and higher hospital costs "

You need to provide the statistical analysis (i.e., indices and values) carried out.
If you do not want to provide this kind of analysis, you should rephrase the sentence accordingly.

Additional comments

Lines 220-237: I appreciate your modifications; you expanded the global perspective properly. However, I would suggest adding also studies from European Countries.
For instance, see this reference, in which you will also find additional information on studies of financial burden in Europe:
Lambiase, C.V., Mansi, G., Salomè, S. et al. The financial burden experienced by families during NICU hospitalization and after discharge: A single center, survey-based study. Eur J Pediatr 183, 903–913 (2024). https://doi.org/10.1007/s00431-023-05352-y

---

## Round 0.3 · accepted · Accept

· Academic Editor

Accept

Thank you for revising your manuscript to address the concerns of the reviewers. Reviewer 2 now recommends acceptance and I am satisfied that the comments of reviewer 1 have been addressed. The manuscript is now ready for publication.

·

Basic reporting

Authors answered to all the requests properly.

Experimental design

Authors answered to all the requests properly.

Validity of the findings

Authors answered to all the requests properly.